# The Effect of Health Education on Adolescents’ Awareness of HPV Infections and Attitudes towards HPV Vaccination in Greece

**DOI:** 10.3390/ijerph19010503

**Published:** 2022-01-03

**Authors:** Ioannis Thanasas, Giagkos Lavranos, Pinelopi Gkogkou, Dimitrios Paraskevis

**Affiliations:** 1Department of Obstetrics & Gynecology, General Hospital in Trikala, 42100 Trikala, Greece; 2Department of Health Sciences, Medical School, European University Cyprus, 2404 Nicosia, Cyprus; G.Lavranos@external.euc.ac.cy; 3Clinical Oncology Department, Norfolk and Norwich University Hospital, Norwich NR4 7UY, Norfolk, UK; PINELOPI.GKOGKOU@nnuh.nhs.uk; 4Department of Hygiene Epidemiology and Medical Statistics, Medical School, National and Kapodistrian University of Athens, 15772 Athens, Greece; dparask@med.uoa.gr

**Keywords:** HPV, HPV vaccines, teenagers, health education, Central Greece

## Abstract

Background: The purpose of this paper is to evaluate the role of health education among young adolescents, regarding their level of knowledge about the HPV and the acceptance of the HPV vaccination, with the aim of increasing vaccination coverage, in Trikala city, mainland of Greece. Methods: This study included high school students from all public and private schools of Trikala city. Questionnaires related to knowledge regarding the HPV infection and HPV vaccination were administered in three phases. In the first phase of the study, the questionnaires were completed by the entire study population. During the second and third phases, the completion of the questionnaires concerned only the population that received the intervention (interactive seminars). The second and third phase questionnaires were completed immediately after the interactive seminar and three months later, respectively. The statistical analysis of the results was performed using IBM SPSS 20.0 statistical program. Results: A total of 434 female students completed the questionnaire (response rate 76.26%). Most participants (66.6%) were females. The questionnaires evaluated the respondents’ awareness of the HPV infection. The results show that the percentage of the participants who stated that they “do not know” what HPV is decreased from 44.4% (first phase), to 1.6% (second phase), and 8.1% (third phase). Similarly, the willingness to accept the HPV vaccine increased from 71% (first phase), to 89.1% (second phase), and 83.5% (third phase). Conclusions: The present study shows that targeted interactive informational interventions in the school environment leads to a statistically significant increase in both the level of knowledge about HPV and the willingness of young adolescent students to be vaccinated against cervical cancer.

## 1. Introduction

The Human Papillomavirus (HPV) is currently the leading cause of cervical cancer. The indisputable finding of a causal link between HPV and cervical cancer has led the scientific community to the decision to create HPV vaccines. In 2006, the quadrivalent vaccine (Gardasil^®^) was first released around the world, and a year later the bivalent vaccine (Cervarix^®^) was approved for circulation. Most recently in 2014, the nine-valent vaccine against HPV (Gardasil 9^®^) was released. The HPV vaccine has been incorporated into the Greek national vaccination program since 2008 and since 2017 it is administered for free only until the age of 16 years of age [1].

In Europe, the HPV vaccination program has been adopted by the majority of the Member States of the European Union [2]. In 2010, it was estimated that the vaccination coverage of the population ranged between 17% and 81% [3]. The average vaccination coverage of Greek adolescents (aged 11–16 years) during the period 2008–2014 was only 8.9%, increasing from 3.2% in 2008 to 17.1% in 2011 [4]. According to the World Health Organization’s declaration [5] and the outcomes of numerous research studies, the target population of the HPV vaccination should be young adolescent students in primary and secondary education [6].

Acceptance of the HPV vaccination is an essential goal of public health practice in every country, including Greece, in order to achieve the vaccination coverage goal of 90% [7]. The aim of this study is to investigate whether the role of health education could possibly enhance the awareness of young adolescents regarding HPV, as well as their acceptance of the HPV vaccination. The absence of many research studies to evaluate educational interventions regarding the sensitization of young adolescents regarding HPV and the vaccination against cervical cancer in Greece and especially in the region of Trikala makes this study very useful.

## 2. Material and Methods

In the school year 2018/2019, an intervention study was conducted on HPV and its prevention, aimed at pre-adolescent students living in cities, towns, and villages of the Prefecture of Trikala in Central Greece.

### 2.1. Study Sample

The participants of the study were students of the 1st Grade Class of Gymnasium (Equivalent to High/Secondary School). The inclusion criteria consisted of pre-adolescent school students, 11–12 years old, attending first-grade secondary schools in the province of Trikala in Central Greece. The study proposed the participation of all first-grade students from seven school groups in the prefecture of Trikala distributed as follows: four schools were selected from the city of Trikala (urban school complexes), two schools from the province of Kalampaka in the prefecture of Trikala (rural school complexes), and the only private school in the prefecture was also selected. The four urban school complexes were categorized into two groups, each of which included two schools: one school was the sample of intervention of the studied population and the other school was the control sample of the studied population. The 1st and 3rd Gymnasiums of Trikala were included in the urban schools, whose students comprised the studied intervention population. The 4th Gymnasium of Trikala and the 9th Gymnasium of Trikala were the control sample of the studied population. Similar to the two rural school complexes in the province of Kalampaka, one (1st Gymnasium of Kalampaka) was the sample of intervention and the other school (2nd Gymnasum of Kalampaka) was the control sample of the studied population. In the only Private Gymnasium of the city of Trikala, half of the students of the 1st grade of Gymnasium were the sample of intervention and the other half the sample of non-intervention—the control of the studied population. The ethical approval to conduct this research was obtained after the submission and approval of the protocol from the Institute of Educational Policy of the Ministry of Education, Research, and Religions (Research and Ethics Committee Decision No. 219048/Δ2/19-12-2018).The anonymity and confidentiality of participants was ensured from the study protocol. Out of 573 questionnaires offered in paper form to the students, 434 completed questionnaires were returned. The largest response was observed in the Private High School (95%), while the 4th Gymnasium of Trikala and the 1st Gymnasium of Kalampaka were those with the lowest participation rates, 68.1% and 68.1%, respectively (Table 1).

### 2.2. Questionnaire

The methodological tool of the study was a questionnaire. The questionnaire was chosen after a systematic bibliographic review of studies with a similar research question. The license to use the questionnaire was obtained after contacting the editorial team [8]. Health professionals reviewed the questions to test the acceptance and feasibility of the questionnaire. To pilot test the questionnaire prior to the beginning of the trial, a health professional was approached that had been identified as being willing to volunteer to use the questionnaire. The data were analyzed quantitatively using descriptive statistics and analysis of its reliability. The aim was to further refine the questionnaires by assessing their acceptability. The final form of the questionnaire consisted of 52 closed–ended questions, which were divided into four main sections. In the first and second sections school data and student demographics were included, respectively. The third section included students’ knowledge, attitude, and behavior toward the HPV infection and their awareness of the relationship between the virus and cervical cancer. The fourth section included students’ knowledge, attitude, and behavior toward the HPV vaccination and cervical cancer. The data collected from the questionnaire were anonymized. No clinical data were collected.

### 2.3. Study Schedule

The survey was conducted during the academic year of 2018–2019. The time period for the study was from January to May 2019. The first phase of the study included the completion of questionnaires by all participants (i.e., the intervention group and control group). The intervention group was randomly selected to be included in the study. Two weeks later, an informational lecture was given during allocated school hours that were agreed with the Head of School. The completion of the questionnaire by all the students took place at the end of the seminar (second phase). The informational lectures to the students took place during working days and hours. The duration of the students’ briefing did not exceed 45 min (Figure 1). The control group was excluded in the phase II and III, as the two aims of the study were to recognize the level of the knowledge in the entire population and secondly to understand whether school-based programs with reliable interventions such as lectures or interactive discussions can increase the understanding of the HPV infection and the acceptance of the HPV vaccination. The educational intervention was delivered face to face using an audio-visual presentation. It was directed at individual groups and included information about the risks and benefits of vaccines; where, how, and when to access vaccine services; who should be vaccinated; or a combination of these topics, as well as information regarding the HPV infection and its signs and symptoms.

### 2.4. Data Collection

The collection of the questionnaires, as well as the informative sessions, was carried out by one author (I.T.) of the study. The survey consisted of anonymous questionnaires. Participant pre-adolescents and their parents were shown subscribed assent and consent documents, respectively, informing them of the study’s aims and objectives, as well as the participant’s right to refuse or terminate their participation in the study at any time with no disadvantages.

### 2.5. Data Statistics

The IBM SPSS (Statistical Package for the Social Sciences) 20.0 statistical program (IBM Corp., Armonk, NY, USA), acquired by IBM, was used to analyze the sample data. Using descriptive methods, the sample was analyzed by taking measurements of the frequency and percentages of responses to all questions. The statistical analysis was blinded to the researchers and conducted independently. A Pearson X^2^ statistical test was used to evaluate the possible dependence between questionnaire variables and students’ knowledge status and behavior towards the HPV. The results between the three phases of the study were compared, both with the analysis of the tables obtained from the extraction of descriptive measures and with the use of the statistical paired *t*-test for dependent samples, in the 95% confidence interval. Through these comparisons, we received the necessary information to enhance students’ awareness about and behavior towards HPV and the vaccination against cervical cancer. Sample with *p*-value = 0.05 were considered statistically significant.

## 3. Results

### 3.1. General Characteristics

A total of 434 students who met the criteria for participation in the study filled the questionnaire, 289 (66.6%) of which were girls and 145 (33.4%) were boys. The vast majority of participants (95.6%) were of Greek nationality. Overall, the general demographic characteristics of the study population are summarized in Table 2. The intervention group consisted of 248 students.

### 3.2. Knowledge about HPV

The analysis of the data revealed that no statistically significant difference was observed in the level of knowledge about HPV based on the sex of the participants (Phase II). On the other hand, a statistically significant difference was observed between the levels of students’ annual family income, their nationalities, and the different residential areas of the participants. Similarly, increased levels of knowledge about the virus were noticed among participants of Greek origin, compared to Albanians or other nationalities, as well as in those living in urban centers compared to those living in villages (Table 3).

### 3.3. Willingness to Receive the HPV Vaccination

Analysis of the data from our research (Table 4) shows that there is a statistically significant difference in the rates of vaccination willingness against HPV based on sex, annual family income, and place of residence. It is worth noting that a much higher rate of willingness to vaccinate was observed in girls than boys. At the same time, students who came from high-income families or urban areas were more likely to be vaccinated, especially compared to the ones coming from the rural area No statistically significant difference was observed in the willingness rates for vaccination based on the nationality of the participants.

### 3.4. Differentiation of the Level of Awareness about HPV and the Attitudestowards Vaccination (Willingness or Unwillingness)

According to the final analysis of the results, there was a rapid change regarding the level of knowledge about HPV between the three phases of the study:

In the first phase of the study, the percentage of participants who stated that “I don’t know” what HPV is was 44.4% (n = 110), while 52.4% of young adolescent students (n = 130) replied that it was a virus. In addition, 38.3% (n = 95) answered that sexual intercourse is the most common mode of transmission.

After the intervention (phase II), participants who stated “I don’t know” what HPV is decreased to 1.6% (n = 4), while 95.6% (n = 237) of those responded it was a virus. At the same time, 95.6% (n = 237) of the participants answered that sexual intercourse is the most common mode of transmission.

Three months after the intervention (phase III), participants who replied “I don’t know” what HPV is had a percentage of 8.1% (n = 20), while 88.3% (n = 219) stated that it is a virus. Furthermore, 86.3% (n = 214) of the participants indicated that the most common way of viral transmission is sexual intercourse.

Moreover, a change in the responses of the studied population was noticed regarding the willingness to receive the HPV vaccination among young adolescent students. Specifically, in phase 1, 71% (n = 176) of the sample were willing to be vaccinated, in phase 2 (immediately after the informative intervention) this percentage rose to 89.1% (n = 221), and finally, in phase 3 (three months after the informative intervention) the willingness rate for HPV vaccination was 83.5% (n = 207).

From the comparative analysis of the data of our sample, we observed that in the third phase of the study the percentage of students (88.3%) who answered that “HPV is a virus” was statistically significantly different from the results of the first phase (52.4%), but slightly reduced compared to the second phase of the study (95.6%).

In the second phase of the study, the percentage appears to have increased, but this was without statistical significance in the percentage of those who stated that “they do not know” what HPV is (8.1% in phase III versus 1.6% in phase II). Overall, it is estimated that the level of knowledge of our population in relation to the first phase of sampling is statistically significantly different for the better. When asked about the ways in which the virus is transmitted, 86.3% answered “through sexual intercourse”. This percentage is slightly reduced compared with the second phase of the study (i.e., by 9.3%), but rapidly increased compared to the first phase of the study (86.3% vs. 38.3%, respectively).

The rate of willingness to be vaccinated against HPV and cervical cancer in young adolescent students in the sample of our study in the third phase remains high (83.5%). This percentage is slightly reduced compared to the second phase of sampling (89.1%), but significantly increased compared to the first phase of the study, which showed only 71.0% of participants were willing to be vaccinated.

Collectively, in the above questions examined across the three phases of the study, it was observed that there is a statistically significant difference in the mean values of the sample responses between the first phase and the third phase. At the same time, however, there was a decrease in rates, both in terms of knowledge about HPV and some questions about vaccination, as well as the willingness of individuals to be vaccinated between the second and third phases of the study, but without a statistically significant differentiation.

## 4. Discussion

In this study, we recorded young adolescents’ levels of awareness about HPV and their willingness to be vaccinated against it. Vaccination against HPV may be proven to be the best strategic option for the country’s health system. In fact, vaccinating girls at a young age, before the initiation of sexual activity, is the preferred strategy [9].

Our data indicate that the level of knowledge about HPV and the vaccination against cervical cancer is not statistically significant between the two sexes. This finding is opposed to findings from other studies. Durosoy et al. showed that women were better informed about the effects of HPV infection, which was presumed to be related to the campaigns aimed mainly at female population rather than at males [10]. In recent studies, it has been revealed that female students had a higher level of knowledge and a more positive attitude towards HPV than male participants [6,11].

The present study revealed that the residence area of young adolescent students is strongly related to their levels of knowledge about HPV and their willingness to be vaccinated against cervical cancer. It is concluded that students living in urban centers are more aware of HPV and HPV vaccination than students living in rural areas. Students residing in villages tend to be unwilling to be vaccinated against cervical cancer. This is a unique finding, as the only study that comments on the different between areas of the same country is Durusoy et al. That study concluded that participants from Turkey’s western region were more willing to accept and be included in a preventive vaccination program than the residents of the mainland [10]. Studies from Hungary and the Netherlands have also demonstrated a higher level of awareness about HPV, compared with Germany, suggesting that the knowledge about the virus and HPV vaccination is a multifactorial issue [12,13].

According to the results of our study, we concluded that annual family income is statistically significantly related to the level of knowledge about HPV and the willingness to undergo vaccination. Families with higher incomes usually offer better education to their children, which resulted in higher levels of awareness about HPV. This is perhaps the case, because these parents are themselves better educated and so are able to educate their children better. Similar outcomes were observed by Durosoy et al. [10]. A recent study found that the education level of parents was a slightly more important factor than their socioeconomic level, in terms of acceptance of the HPV vaccination [14]. In contrast, a recent study from Slovenia indicated that the lowest vaccination rates were not statistically significantly correlated with lower average family incomes, nor was there any relationship between higher education ratios and the immunization rate of the participants in the study [15].

The marital status of the parents of the participants in this study was not statistically significantly related to the level of knowledge about HPV and the willingness to be vaccinated against cervical cancer. Similarly, in a previous study, the authors concluded that marital status was not statistically significantly related to the acceptance of the HPV vaccination [16]. In contrast, a recent study indicated that the participants’ family status, along with other socio-demographic factors, and the age of the unvaccinated daughter, appeared to act as barriers to the decision to vaccinate against cervical cancer [17].

Analysis of the data of this study reveals that nationality is statistically significantly related to the level of knowledge about HPV and the willingness to vaccinate. Greek students appear, in our study, to be much more informed about the virus than students of Albanian or other nationalities. Nationality status might, in this instance, be confounded by economic and educational status. However, it is necessary to point out the observed disproportion of nationalities among the participants, 95.6% are of Greek nationality, which does not allow us to statistically validate this result. Similarly, a study from Italy indicated lower vaccination coverage rates against HPV in foreigners compared to Italian citizens [18]. Furthermore, other studies involving female adolescents of national minorities from degraded areas, found that the rates of HPV vaccination were lower compared to those of privileged ones [19,20].

The targeted informative intervention showed a statistically significant increase in both the level of knowledge about HPV and the infection as well as in the adolescents’ willingness to be vaccinated against HPV. This was the main result of our research study. Ignorance about the HPV infection and its implications, as well as the benefit of HPV preventive vaccination, is still high among adolescents. The way to improve their knowledge about HPV and the implications of the HPV infection is to provide information through the framework of compulsory schooling, primary health care, and the development of informative interactive interventions. The knowledge about and the perceptible susceptibility to the HPV infection and HPV-related diseases among adolescents demonstrate the need for a well-designed training program to bridge the gap of information about HPV and to encourage acceptance of the HPV vaccine. Implementation of intervention materials, particularly the communication techniques, can improve adolescent HPV vaccination acceptance.

The knowledge about and receptive susceptibility to the HPV infection and to HPV-related health issues demonstrate the urgent need for a well-designed training program, in order to bridge the gap between HPV awareness and vaccine acceptance [21].

Previous studies have also indicated that informing young women is an important factor in shaping positive thinking and perception towards the vaccination against HPV [10,22,23]. On the other hand, as has been shown in other studies, the participation and completion of a preventive HPV vaccination program is not directly correlated with the level of awareness of vaccinated women [24,25].

### Limitations of the Study

The main limitations of the present study are the representativeness of the study population. Furthermore, selection bias could have an impact on the participation of school students whose parents were more prone to vaccine compliance. The role of the physician (I.T.) who provided counselling and recommendations for the HPV vaccine target populations may have played a crucial role in the study, biasing the result towards vaccine compliance. The control group was only measured at the baseline and not at other time points, due to the fact that we needed to monitor the intervention (health education). Since few school students refused to participate in the study, the reasons for refusal were not investigated.

## 5. Conclusions

Young adolescent students are poorly informed about HPV and the HPV vaccination, which can be a significant obstacle to the necessary increase needed in the vaccination coverage of the population. The use of interactive lecturing interventions, provided to young adolescent students in the school environment, could contribute in a statistically significant way to the increase in both the level of knowledge about HPV and the infection caused by it, as well as in the willingness of participants to be vaccinated against HPV.

At the same time, the implementation of health education programs in schools, under the responsibility of the State, can be an important measure in an effort to reduce the occurrence of cervical cancer. Sex education of young girls and boys in order to gain a good amount of knowledge concerning issues such as sex life, family planning, proper use and effectiveness of contraceptive methods, and good hygiene, reduces the exposure of the population to predisposing risk factors for developing HPV.

## Figures and Tables

**Figure 1 ijerph-19-00503-f001:**
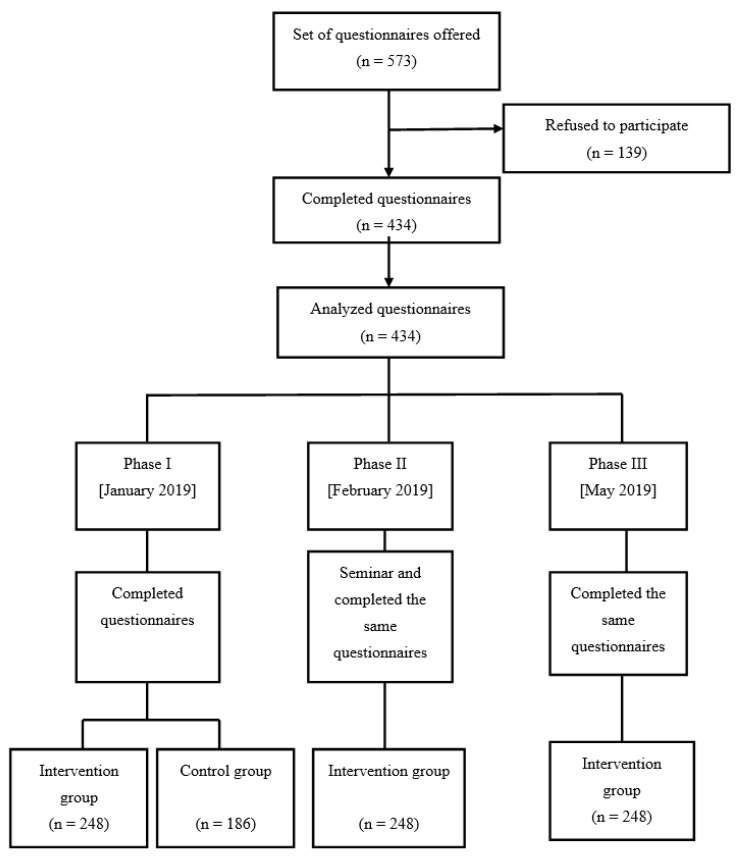
The flow diagram of the study.

**Table 1 ijerph-19-00503-t001:** Phase I: Response rate of all participants (intervention and control groups) to the questionnaires per school.

Study Population			
Control Group	Specimen	Intervention Group	Specimen
4th Gymnasium of Trikala	62/91 (68.1%)	1st Gymnasium of Trikala	95/119 (79.8%)
9th Gymnasium of Trikala	47/65 (72.3%)	3rd Gymnasium of Trikala	80/98 (81.6%)
2th Gymnasium of Kalampaka	51/74 (68.9%)	1st Gymnasium of Kalampaka	45/66 (68.1%)
Private Gymnasium	57/60 (95%)	Private Gymnasium	57/60 (95%)

**Table 2 ijerph-19-00503-t002:** Participants (intervention and control group) characteristics at the baseline (Phase I) (n = 434).

Demographic Characteristics	Total Sample (n = 434)	Percentage (%)
**Sex**		
Male	145	33.4%
Female	289	66.6%
**Place of residence**		
Big City	181	41.7%
Small Town	185	42.6%
Village	68	15.7%
**Nationality**		
Greek	415	95.6%
Albanian	13	3%
Other	4	1.4%
**Religion**		
Christian Orthodox	422	97.2%
Catholic	4	0.9%
Other	8	1.8%
**Parental Marital Status**		
Married	412	94.9%
Divorced	22	5.1%
**Monthly Family Income**		
High	90	22.8%
Medium	292	64.7%
Low	52	12.2%
**Annual Family Income**		
High	99	22.8%
Medium	282	64.9%
Low	53	12.2%

**Table 3 ijerph-19-00503-t003:** Phase II: demographic characteristics of the intervention group correlated with participants’ level of knowledge about HPV (n = 248).

Demographic Characteristics	Level of Knowledge about HPV		
	n (%)	“I Do Not Know” n (%)	χ^2^ Test
**Total**	248		
Sex			*p* = 0.466
Male	91 (36.7%)	37 (40.7%)	
Female	157 (63.3%)	73 (46.5%)	
**Annual Income**			*p* < 0.01
High	59 (23.8%)	11 (18.6%)	
Medium	146 (58.9%)	59 (40.4%)	
Low	43 (17.3%)	40 (93%)	
**Nationality**			*p* < 0.01
Greek	240 (96.8%)	104 (43.3%)	
Albanian	2 (0.8%)	0 (0.00%)	
Other	6 (2.4%)	6 (100%)	
**Place of Residence**			*p* < 0.01
Big City	132 (53.2%)	64 (48.5%)	
Small Town	77 (31.0%)	13 (16.9%)	
Village	39 (15.7%)	33 (84.6%)	

**Table 4 ijerph-19-00503-t004:** Phase II: demographic characteristics of the intervention group correlated with willingness to receive the HPV vaccination (n = 248).

Demographic Characteristics	Willingness to Receive the HPV Vaccination		
	n (%)	“Willing” n (%)	χ^2^ Test
**Total**	248		
Sex			*p* = 0.028
Male	91 (36.7%)	57 (62.6%)	
Female	157 (63.3%)	119 (75.8%)	
**Annual Income**			*p* < 0.01
High	59 (23.8%)	46 (78.0%)	
Medium	146 (58.9%)	120 (82.2%)	
Low	43 (17.3%)	10 (23.3%)	
**Nationality**			*p* = 0.233
Greek	240 (96.8%)	169 (70.4%)	
Albanian	2 (0.8%)	1 (50.0%)	
Other	6 (2.4%)	6 (100%)	
**Place of Residence**			*p* < 0.01
Big City	132 (53.2%)	101 (76.5%)	
Small Town	77 (31.0%)	62 (80.5%)	
Village	39 (15.7%)	13 (33.3%)	

## Data Availability

The datasets used and/or analysed during the current study are available from the corresponding author upon reasonable request.

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
