# Peer review of "The Effect of Health Education on Adolescents’ Awareness of HPV Infections and Attitudes towards HPV Vaccination in Greece"

_ijerph, 2022, doi:10.3390/ijerph19010503_

Round 1

Reviewer 1 Report

I read the manuscript with great interest, HPV vaccination is an important public health problem and education among adolescents plays an important role.

However, the authors did not efficiently point out what is the element of novelty, what was the purpose of the study, and how this was accomplished? Was it an interventional study to prove that a specific educational intervention would increase HPV vaccine acceptance? If yes (although not appropriately stated), than the major flaw of the study would be the absence of results post-intervention in the “control” group

The only paragraph in the Results section referring to the above (inferred) objective would be the last paragraph, from which we find  the willingness for HPV vaccination among young adolescent students raised from 71% (n=176) of the sample in phase 1, to 89.1% (n=221)in phase 2 (immediately after the informative intervention), and to 83.5% (n=207) in phase 3 (three months after the informative intervention).There is no proof of a cause-effect relationship in the absence of a control group. The same applies to the level of knowledge about the HPV – how do we know that the increased level of knowledge is the consequence of the informative intervention, when given the same questions on HPV, 3 months later, without a control group?

Some point by point observations:

  •  
  • The aim of the study is not clearly stated
  • Study design inappropriately described: “A cross-sectional survey about HPV and its prevention, addressed to pre[1]adolescents and parents of pre-adolescents, was conducted during the 2018/2019 school year”
    • If this was meant to be an interventional study, this should be mentioned in the first Method section
    • If is only a descriptive study through a cross-sectional survey, than why do they speak about “intervention” and “control” groups
  • Other methodological flaws
    • Not clear how students and parents answers were treated? It looks like they were treated the same from the Method Section, but from Results section we understand that “an informational lecture was given to all participants (students)”. what does mean all participants?? Do they include control group? If not, on what bases some had the intervention, others not?
    • - The flow diagram (which might have clarified the above questions, does not include numbers, either)
    • How was the splitting between intervention and control group made?
    • What is the relevance of a “control” group, if not testing of this group is available after intervention?

    • Table 1 – reorganize it on intervention vs control group columns
    • Table 2 with socio-demographic characteristics, does not make any difference between groups
    • How was the intervention group selected?
    • Demographic characteristics of the so called control group are lacking

As the authors themselves pointed out, The above conclusion was drawn from the systematic review that our research team had performed – which would be, than,  the new data that the current research bring to light?

Author Response

Dear Reviewers ,

Please accept for re-submission the manuscript entitled ‘Effect of health education in adolescents' awareness of HPV infections and attitudes towards HPV vaccination in Greece’

We are grateful for the all the comments that both the reviewers took time to address to ourselves.

I would like to confirm that to our knowledge all the comments, observation and recommendations were addressed in the manuscript with the form of tracked changes.

We would be grateful if you please consider this manuscript for publication to your journal.

Looking forward to hearing from you regarding your decision.

Please see the attachments. I have attached the revised article 

Kind Regards

G. Thanasas and co-authors

Reviewer 2 Report

This study provides further insight into the multifactorial causes affecting the HPV vaccination programme in Greece and how the vaccination rate could be increased. However, there are data in the manuscript that are important to analyse. I attach recommendations for improving it. However, it will be difficult to specify each of them because the manuscript does not have line numbering (which is provided in the initial template):

  • The first paragraph of the Introduction: In the last line, consider changing "will be" to "has been".
  • Please consider moving the first two lines of the second paragraph of the introduction before explaining the situation in Greece. The idea is to discuss the European level first and then the national level.
  • Last paragraph of the Introduction, line 3: Please delete "with the use of questionnaires". This is already detailed in the methodology.
  • First paragraph of Material and Methods: If this is a cross-sectional study, why has the completed STROBE checklist for cross-sectional studies not been used and included as an appendix or supplementary material?
  • Second paragraph of Material and Methods, line 9: Please detail whether the questionnaire was offered in paper or online.
  • Table 1: The text of the statement "Study Population" is not aligned with the other statements in the table.
  • Third paragraph of Material and Methods, lines 4-5: Indicate, if available, results on the acceptability and feasibility of the developed questionnaire.
  • Third paragraph of Material and Methods, lines 5-6: Please include the questionnaire as an appendix or supplementary material, as it makes it difficult to interpret some of the results of the manuscript.
  • Third paragraph of Material and Methods, lines 11-12: The term "personal data" is rather unspecific, as socio-demographic data have been collected. Consider replacing the term with "clinical data".
  • Fourth paragraph of Material and Methods, lines 6-8: In my opinion, these lines are poorly expressed. It seems that only the intervention group did the questionnaire in the first phase. It could be shortened to mean that the first phase was conducted to collect data to establish the baseline.
  • Fourth paragraph of Material and Methods, lines 11-12: Some more information is needed on this part of the text: Who conducted the intervention (authors, other health professionals...)? How long did each session last? Were physical materials such as leaflets used?
  • Figure 1: (1) Please delete pilcrows. (2) Indicate the months in which each phase was carried out.
  • Fifth paragraph of Material and Methods, line 2: Please replace "by the author" with "by one author".
  • Sixth paragraph of Material and Methods: Was the statistical analysis blinded to the researchers and conducted independently?
  • Sixth paragraph of Material and Methods, line 1: There is a change of font size in "IBM SPSS (Statistical Package for the Social Sciences)". This happens again in other parts of the manuscript (Discussion and Acknowledgements).
  • Sixth paragraph of Material and Methods, line 3: Apart from frequencies, percentages were also used.
  • First paragraph of Results, lines 4-6: This text could be simplified by simply stating that the intervention group consisted of 248 students.
  • Table 2: It is necessary to change "Gender" to "Sex" as the biological (anatomical) dimension of the variable is being studied, not the psychological and identity dimensions. This should be taken into account whenever gender is mentioned in the text.
  • Second paragraph of Results, lines 1-2: Please indicate the phase at which you are claiming this non-significant difference (Phase II).
  • Second paragraph of Results, lines 5-6: Delete this text, as results are being interpreted. Also, this is a comment that is already mentioned in the Discussion.
  • Table 3: (1) p-value is never equal to 0. Please consider replacing "p=0.00" with "p<0.01". (2) Align "Place of Resident" data appropriately.
  • Third paragraph of Results, lines 5-6: Consider reordering the text: "...come from high-income families or urban areas are more likely to be vaccinated, especially comparing to the ones coming from the rural area".
  • Fourth paragraph of Results (phases 1-3 and "moreover..."): (1) It is difficult to follow the results because not all the answers of the intervention group are shown. (2) It would be interesting to perform a hypothesis test between the IG (comparing phases 1-2 and 2-3) and IG-CG (phases 2 and 1 respectively).
  • First paragraph of Discussion, line 1: Include the letter "r" in "ecorded".
  • First paragraph of Discussion, lines 3-4: Are there any Greek studies that report the mean and standard deviation of the age of initiation of sexual activity in Greek adolescents?
  • Second paragraph of Discussion, line 3: Please add a full stop (.) after "al". This detail is repeated in the following paragraphs.
  • Fourth paragraph of Discussion, line 4: There is a typo in "children".
  • References: (1) Journal names and book titles should be in italics. (2) The year of publication should be in bold type. (3) Between the journal name and the year of publication there should not be a full stop (except if the journal name is abbreviated).
  • Reference number 2: Please add the data for "(accessed on Day Month Year)" after the URL.
  • Reference numbers 6, 17, 18, and 24: Add a space before the journal name.
  • Reference number 11: Please change the Spanish title to Portuguese (as the study is written in Portuguese). Also, this title in Portuguese should be in brackets.

I hope my comments will help you to improve the manuscript.

Best regards.

Author Response

Dear Editor,

Please accept for re-submission the manuscript entitled ‘Effect of health education in adolescents' awareness of HPV infections and attitudes towards HPV vaccination in Greece’

We are grateful for the all the comments that both the reviewers took time to address to ourselves.

I would like to confirm that to our knowledge all the comments, observation and recommendations were addressed in the manuscript with the form of tracked changes.

We would be grateful if you please consider this manuscript for publication to your journal.

Looking forward to hearing from you regarding your decision.

Please see the attachment. I have attached the revised article but as I can attach only one file, it is not possible to attach the questionnaire. I will send the questionnaire attached to the editor.

Kind Regards

G. Thanasas and co-authors

Round 2

Reviewer 1 Report

In their  revision of the manuscript, the authors have, in my opinion, satisfactorily addressed the issues raised, therefore I recommend the paper for publication.

Author Response

Dear Reviewer

Thank you very much for your comments that contributed significantly to the improvement of the article!

Happy holidays!

Best Regards

Ioannis Thanasas

Reviewer 2 Report

Thank you for your efforts to improve the content of the text. After reading the changes, I suggest some necessary modifications:

  • First paragraph on page 4: The comment on the baseline seems to contradict the text (U16). I suggest deleting "in order to collect data to establish the baseline" to avoid confusion between the first and second phase.
  • First paragraph on page 4: Please change "forty – five minutes – that is of, – one teaching hour" to "45 minutes".
  • Figure 1: The flow chart is distorted at the beginning.
  • First paragraph of the "Data Statistics": Please add comment U23 to the text. In lines 2-3, change "We analyzed the sample" to "The sample was analyzed".
  • First paragraph of the "Knowledge about HPV": Please add a full stop after "(Phase II)".
  • Tables 3 and 4: Please unify the width of the table borders.
  • First paragraph of the "Willingness for HPV vaccination", line 2 | Table 4 | Second paragraph of the Discussion: Please replace "gender" with "sex".
  • Graphs 1-3: Please remove them, as the newly added text is sufficiently clarifying.
  • Discussion, line 1: Change "ecorrded" to "recorded".
  • U24, U52 and U53 comments: I believe the font size remains different from the rest of the text (this time it is smaller).
  • Reference number 6: Please add a space before the year of publication.
  • Reference number 11: Please add the English title before the first bracket.

Best regards.

Author Response

Dear Reviewer

Thank you very much for your comments.

I made all the changes you suggested. Thank you very much for your help in improving the article.

I look forward to your reply!

Happy holidays!

Best Regards,

Ioannis Thanasas
